# Lossy Compression of Single-channel Noisy Images by Modern Coders

**Sergii Kryvenko** [1] , **Vladimir Lukin** [1,*] and **Benoit Vozel** [2]

1   Department of Information and Communication Technologies, National Aerospace University,
    61070 Kharkiv, Ukraine; krivenkos@ieee.org
2   CNRS, IETR—UMR 6164, University of Rennes, F-22305 Lannion, France; benoit.vozel@univ-rennes1.fr
*   Correspondence: v.lukin@khai.edu

**Abstract:** Lossy compression of remote-sensing images is a typical stage in their processing chain. In design or selection of methods for lossy compression, it is commonly assumed that images are noise-free. Meanwhile, there are many practical situations where an image or a set of its components are noisy. This fact needs to be taken into account since noise presence leads to specific effects in lossy compressed data. The main effect is the possible existence of the optimal operation point (OOP) shown for JPEG, JPEG2000, some coders based on the discrete cosine transform (DCT), and the better portable graphics (BPG) encoder. However, the performance of such modern coders as AVIF and HEIF with application to noisy images has not been studied yet. In this paper, analysis is carried out for the case of additive white Gaussian noise. We demonstrate that OOP can exist for AVIF and HEIF and the performance characteristics in it are quite similar to those for the BPG encoder. OOP exists with a higher probability for images of simpler structure and/or high-intensity noise, and this takes place according to different metrics including visual quality ones. The problems of providing lossy compression by AVIF or HEIF are shown and an initial solution is proposed. Examples for test and real-life remote-sensing images are presented.

**Keywords:** image lossy compression; better portable graphics; optimal operation point; AVIF and HEIF; additive white noise

## 1. Introduction

The number and volume of remote-sensing (RS) images are rapidly increasing. They are widely employed in ecological monitoring, military, agriculture, and other fields [1–4]. To be useful, RS data often have to be processed and offered to customers quickly. This explains the need for their efficient compression [5].

Lossless compression [6] does not introduce distortions into data but is characterized by a relatively small compression ratio (CR) that can be inappropriate for practice. In contrast, lossy compression approaches are possible [7–9] that allow introducing certain distortions [10,11], but the CR for them is usually considerably larger and can be varied. A problem that arises then is to produce a reasonable compromise between the CR and RS data quality [10–13].

In this sense, it is usually assumed that a larger CR results in larger distortions [12,14,15]. In other words, it is supposed that rate/distortion curves (dependence of a parameter characterizing quality on a parameter that controls compression (PCC), i.e., allows varying of the CR) are monotonous. In most practical situations, the aforementioned assumptions are valid. The examples are the dependence of the peak signal-to-noise ratio (PSNR) on the quality factor for JPEG (that monotonically increases [16]) or the dependence of the PSNR on the parameter Q (that serves as the PCC for the better portable graphics (BPG) encoder [17]) that monotonically decreases (see the plots in Figure 2 in [18]). Such behavior allows us to find a trade-off between the CR and distortions quite easily. Note that distortions can be characterized by conventional metrics such as the PSNR, visual quality metrics [15] such as,

e.g., SSIM [14], and, indirectly, by their influence on the classification accuracy of compressed RS data [11,13].

The aforementioned properties and tendencies in lossy compression take place if the original images are noise-free or, at least, the noise intensity is low (the noise is practically invisible). However, RS images (or, at least, some of their components) are not always noise-free. There are applications (types of RS images) for which images are always noisy, e.g., radar images [10,19,20]. There are noisy (junk) component images in hyperspectral [21,22] and multispectral [11] RS data. Night light images [23] are quite noisy as well.

The lossy compression of noisy images has specific features discovered almost three decades ago [24–27]. These features were shown first for JPEG [28] and then for wavelet-based coders [26]. A noise filtering effect that takes place due to lossy compression was demonstrated. This effect can result in an optimal operation point (OOP) that is not always observed, under certain properties of an image to be compressed and noise present in this image. In fact, the OOP is a value of a used PCC for a given coder for which a minimal "distance" (according to a considered metric) between the compressed and noise-free images takes place. For the description of this distance, it is possible to apply both traditional metrics (e.g., PSNR) and visual quality metrics such as PSNR-HVS-M [29] or MS-SSIM [30]. In most cases, the OOP corresponds to a maximal value of a used similarity measure (PSNR, PSNR-HVS-M, MS-SSIM), although it might also relate to minimal value (e.g., for MSE).

After the design of new encoders, research in the area of the lossy compression of noisy images was continued. The possible existence of an OOP was shown for JPEG2000 [31] in [32]. In [32–34], the potential existence of an OOP was demonstrated for the coders AGU [35] and adaptive DCT (ADCT) [36] based on the discrete cosine transform (DCT) in fixed [35] and adaptively determined [36] size. An OOP also might exist in the lossy compression of noisy images by the BPG encoder [18,37]. The possible existence of an OOP has been recently shown for lossy compression based on the discrete atomic transform [38] and tensor decomposition [22]. Also note that modern compression techniques are often designed to carry out noise removal and compression simultaneously [39,40].

Thus, it can be concluded that the possible existence of an OOP is an inherent property of lossy compression applied to noisy images including the cases of signal-dependent noise [26,41]. An attractive advantage of the OOP is that, if it exists for a given image and a coder used, then it is reasonable to compress this image in the OOP or its nearest neighborhood since this provides the best achieved quality of the compressed image and quite a large CR simultaneously [18,41].

Recall here that the BPG encoder was proposed about ten years ago as a part of the video format HEVC (High Efficiency Video Coding) and has already gained popularity [7,42]. Meanwhile, recently two other encoders, AV1 Image File Format (AVIF) [43] and High Efficiency Image File Format (HEIF) [44], have been proposed. To the best of our knowledge, their performance has not been studied for the compression of noisy and RS images. At the same time, their performance for noise-free images has been compared to other coders and shown to have certain benefits [45,46]. Thus, our first goal is to check whether or not an OOP is possible for AVIF and HEIF encoders applied to noisy images and to compare the coders' performance to other known counterparts such as JPEG and the ADCT and BPG encoders.

We have already mentioned the advantages of lossy compression in the OOP if it exists. However, since noise-free images are absent, it is impossible to determine the OOP by calculating the metric values for compressed and noise-free images using a set of PCC values. Several solutions have been already proposed for the JPEG, JPEG2000, AGU, ADCT, and BPG encoders. First, it has been shown for some coders that, in the OOP, the MSE calculated between the compressed and original (noisy) images is approximately equal to the equivalent variance of the noise in the original image [32] (here we assume that the noise type and statistics are a priori known or pre-estimated with high accuracy [47,48]). This allows us to find such a PCC value that this condition is satisfied. The problem is that

such a procedure for reaching the OOP requires multiple compression/decompression of a considered image. This might take too much time, especially if the number of required iterations is large enough.

Second, it has been shown for the AGU, ADCT, and BPG coders that the OOP can be reached considerably faster (without iterations) using simple expressions that connect the optimal PCC (the quantization step (QS) for the AGU and ADCT coders and the parameter Q for the BPG encoder) with the noise equivalent variance. However, if an OOP exists for AVIF and HEIF, it is unclear how to reach it for a given image and noise variance. To study this question is the second goal of this paper.

We carry out our study for the case of single-channel images corrupted by zero mean additive white Gaussian noise (AWGN). The reasons for this are the following. First, AWGN is considered to be a starting point in research dealing with many aspects of image processing [49] including the blind estimation of noise variance [50], denoising [51,52], and lossy compression [53]. Second, if the noise in original image is signal-dependent, it can be converted to additive by the corresponding variance-stabilizing transform [41,54]. Then, lossy compression is applied after the direct transform and the corresponding inverse transform is applied after image decompression (the corresponding example is considered in Section 4 of this paper). Third, we consider the single-channel image compression because component-wise compression is sometimes applied to multichannel RS images and, also, the preliminary analysis of component-wise compression allows us to understand what to do for a multichannel case [41,53]. Note that the joint compression of several component images usually provides better results than component-wise compression. However, to exploit this property, component-wise images should be properly pre-processed (prepared) for three-dimensional compression and the corresponding versions of coders have to exist (be designed).

The structure of the paper is the following. The image/noise model and the used metrics are considered in Section 2. Explanations of basic dependencies and results are given in Section 3. It also deals with the main analysis of the dependence of the analyzed metrics on the PCC for five encoders. Practical aspects of reaching OOP for AVIF and HEIF and discussions are given in Section 4. Finally, the conclusions are presented.

## 2. Materials and Methods

Assuming that our noise follows the AWGN model, an observed noisy image can be presented as

$$I_{ij}^{n} = I_{ij}^{true} + n_{ij}, \tag{1}$$

where $I_{ij}^{true}$, $i = 1, .., I_{Im}, j = 1, .., J_{Im}$ denotes the true or noise-free image, $n_{ijk}$ is AWGN in the ij-th pixel, and $I_{Im}$ and $J_{Im}$ define the image size. AWGN variance is supposed to be equal to $\sigma^2$ and known or accurately pre-estimated in advance. Examples of noisy component images from AVIRIS hyperspectral data are presented in Figure 1. Analysis carried out in [41,48] has shown that the observed noise can be both close to purely additive and signal-dependent. Note that the input PSNR for such component images for which noise is clearly visible usually varies from 20 to 35 dB for hyperspectral data and can be even smaller for radar images.

It is well-known that image processing efficiency considerably depends on image complexity where, for noise-free images, their complexity can be described by entropy. Because of this, we have carried out the main analysis for the test image Frisco (Figure 2a), which is of simple structure, and the image Diego (Figure 2b), which contains a lot of texture and small details. Both images have been taken from the database SIPI (https://sipi.usc.edu/database/database.php?volume=aerials, accessed on 1 March 2024) where they are presented as color images in tiff format. Their conversion to grayscale form has been conducted.

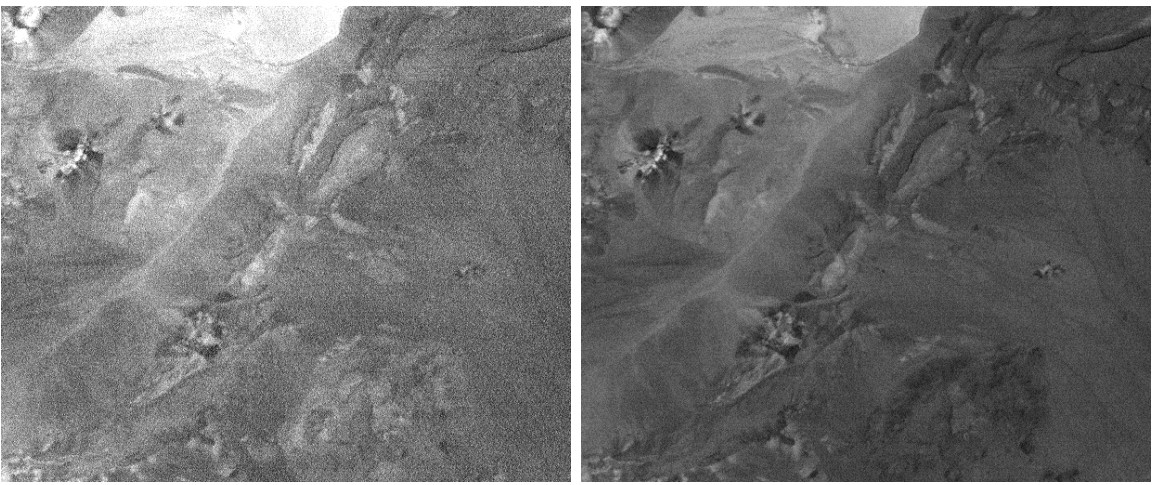

**Figure 1.** Two examples of noisy component images in hyperspectral AVIRIS data.

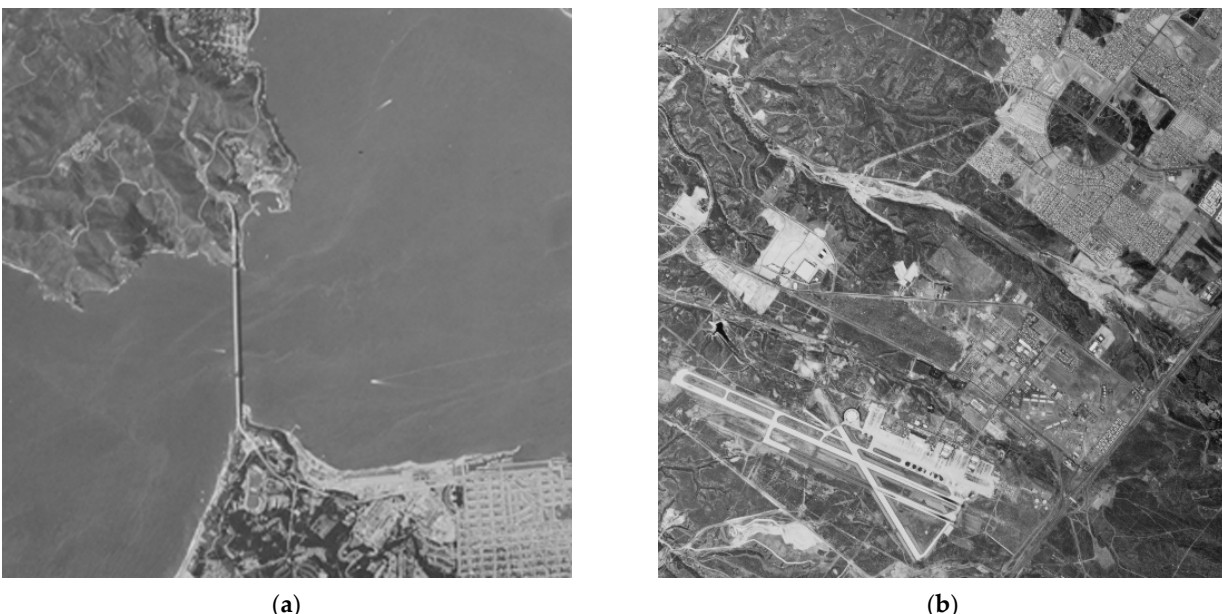

(**a**)                            (**b**)

**Figure 2.** Test single-channel images Frisco (**a**) and Diego (**b**), both $512 \times 512$ pixels.

The quality of the original noisy image represented as 8-bit data can be characterized by

$$\mathrm{PSNR^n} = 10\log_{10}\left(\frac{255^2}{\sigma^2}\right). \tag{2}$$

A conventional rate/distortion curve is the dependence of a metric $\mathrm{PSNR_{nc}}$ calculated between an original (in our case, noisy) and compressed image on a PCC for a given coder. Figure 3 presents such dependencies for all five coders for two images corrupted by AWGN with a variance equal to 100. Before their analysis, recall the following. The horizontal axis relates to PCCs that are different for different coders and vary in different limits. For the BPG encoder, Q serves as the PCC; it can only be an integer and varies from 1 to 51. A larger Q relates to a larger CR. For JPEG, AVIF, and HEIF, the quality factor (QF) serves as the PCC, it can be only integer and a smaller QF relates to worse quality. Finally, QS serves as the PCC for the ADCT coder, and it can be any positive value. Here, we analyze QS from 0 to 100 to be within the same range with other PCCs where a larger QS relates to a worse quality (larger distortions).

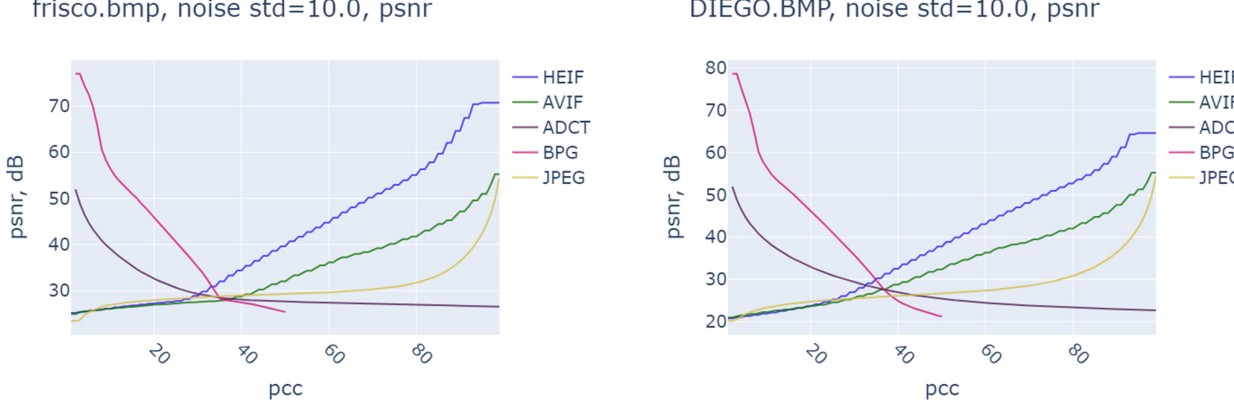

**Figure 3.** Traditional rate/distortion curves for the test images Frisco (**a**) and Diego (**b**) corrupted by noise subject to compression by the five considered coders, $\sigma^2 = 100$.

Due to different PCCs, the dependencies are decreasing for the BPG and ADCT coders and increasing for the three other encoders. PSNR values vary in wide limits from invisible distortions ($PSNR_{nc} > 35\ldots38$ dB) to quite a low quality of the compressed images ($PSNR_{nc}$ about 25 dB for Frisco and about 20 dB for Diego images). All dependencies are not linear, some of them are smooth whilst the plots for AVIF and HEIF are staircase ones.

Figure 3 does not allow to compare the performance of the considered coders. Hence, we have presented $PSNR_{nc}$ values determined between the original and compressed images as the functions of the CR (Figure 4). The main observations are the following. First, the curves for four modern coders practically coincide for both test images. Only for JPEG are the PSNR values significantly smaller, especially for large compression ratios. Second, for the same CR, $PSNR_{nc}$ values for a simpler structure image are larger than for a textured image (compare, e.g., the data for CR = 20 in Figure 4a,b).

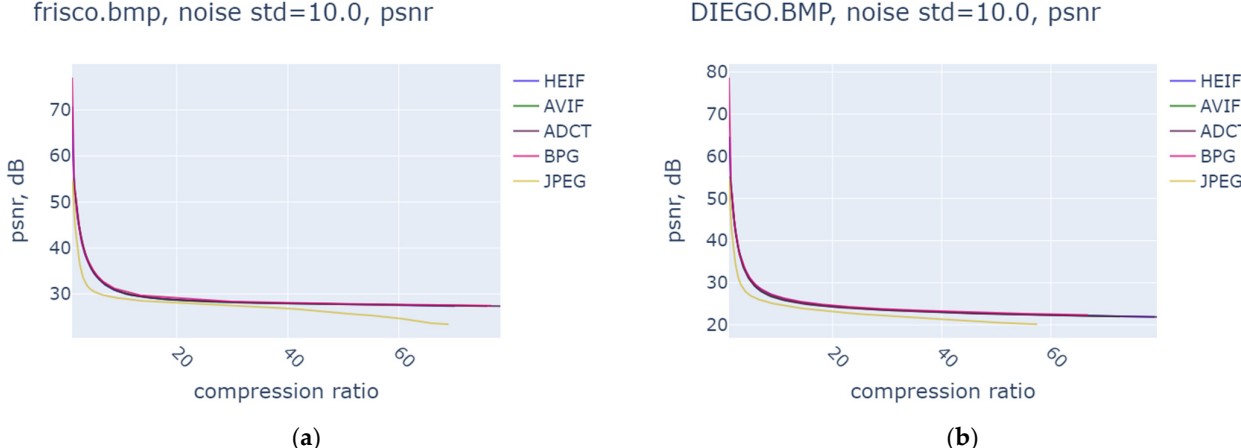

**Figure 4.** Dependencies $PSNR_{nc}(Q)$ for the images Frisco (**a**) and Diego (**b**), $\sigma^2 = 100$.

Meanwhile, for the lossy compression of noisy images, we are more interested in the dependencies of metrics calculated between compressed images and the corresponding noise-free images on the PCC and/or CR. Such dependencies can be obtained only for simulated data where one has a noise-free test image, artificially adds AWGN to it, and compresses it using a set of PCC values to obtain the dependence of the metric $Metr_{tc}$ on the PCC or CR where the index tc corresponds to "true" and "compressed". These dependencies are studied in the next section.

Recall here that, alongside conventional PSNR, it is worth analyzing visual quality metrics that take into account some important features of the human vision system (HVS).

PSNR-HVS-M is expressed in dB; MS-SSIM varies from 0 to 1. Larger values for both metrics correspond to a better visual quality.

## 3. Results

The goals of this section are to study the dependencies of $Metr_{tc}$ on the PCC or CR for different noise intensities as well as to compare the coders' performance. The main attention is paid to OOP existence and performance analysis for different coders (with the main emphasis on AVIF and HEIF) in the neighborhood of the OOP (if it exists).

Dependencies $PSNR_{tc}(PCC)$ for the considered coders for AWGN having $\sigma^2 = 100$ are shown in Figure 5. In agreement with the theory [18,33,37], for the test image Frisco, there are OOPs for the ADCT and BPG coders observed for $QS_{OOP} \approx 4.2\sigma$ and $Q_{OOP} \approx 14.9 + 20\log_{10}(\sigma)$, respectively. The $PSNR_{tc}$ values in these OOPs are the largest and they are 7 dB larger than $PSNR^n = 28.1$ dB. OOPs are also observed for all three other coders where the smallest $PSNR_{OOP}$ is seen for JPEG (about 32 dB). AVIF and HEIF produce approximately the same $PSNR_{OOP}$, this is seen for different values of $QF_{OOP}$. For the test image Diego (Figure 5b), OOPs are not observed (there are no maxima since all dependencies are either monotonically increasing or decreasing) and this shows the dependence of OOP existence on image complexity.

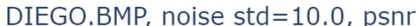

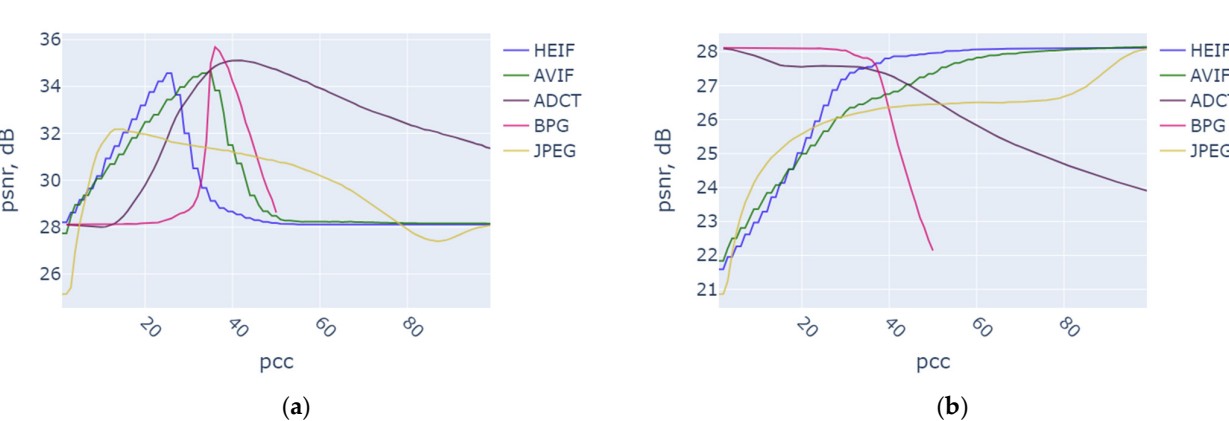

**Figure 5.** Dependencies $PSNR_{tc}(PCC)$ for the images Frisco (**a**) and Diego (**b**), $\sigma^2 = 100$.

Figures 6 and 7 demonstrate the dependencies $PSNR\text{-}HVS\text{-}M_{tc}(PCC)$ and $MS\text{-}SSIM_{tc}(PCC)$ for the same noise variance. For the metric PSNR-HVS-M (Figure 6), OOPs are observed for the AVIF, HEIF, ADCT, and BPG encoders where the best results are achieved for the two latter ones. Formally, an OOP is not observed for JPEG although the dependence has a local maximum for the same QF as in Figure 5. For the test image Diego, no OOPs are seen.

According to the metric $MS\text{-}SSIM_{tc}$ (Figure 7), OOPs are observed for all five coders for the test image Frisco (Figure 7a) and they are absent for the test image Diego (Figure 7b). $MS\text{-}SSIM_{OOP}$ is the largest for the BPG encoder. Note that the OOPs for different metrics for a given test image and noise variance practically coincide. For example, for the image Frisco, $\sigma^2 = 100$, $QF_{OOP}$ for HEIF is about 24 for the curves in Figures 5a, 6a and 7a. This is a positive practical aspect. Really, if there is a way to determine the OOP, it is simultaneously determined for both conventional and visual quality metrics. Recall that there are coders (e.g., AGU [36]) for which the OOP values according to different quality metrics for a given image and noise intensity do not coincide.

Let us see what happens for other values of noise variance. Data for $\sigma^2 = 25$ are represented in Figure 8. Again, OOPs are observed for the test image Frisco and they are absent for the test image Diego, showing that image complexity plays a key role in OOP existence.

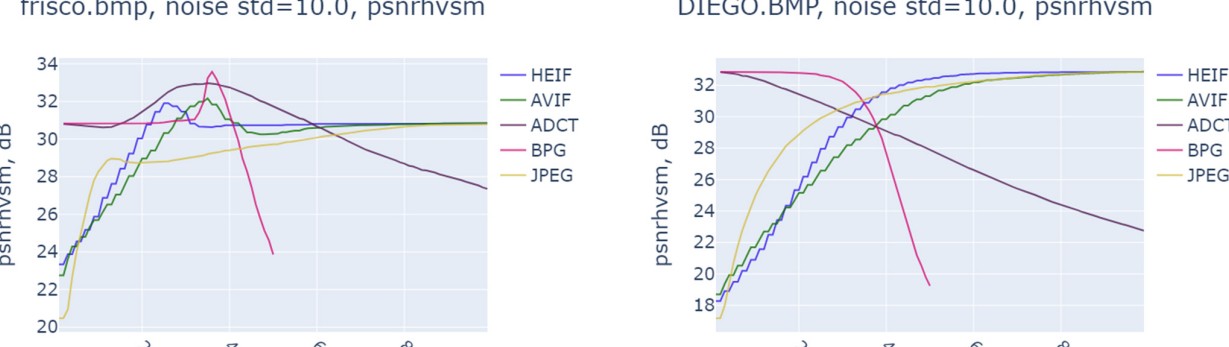

**Figure 6.** Dependencies PSNR-HVS-M$_{tc}$(PCC) for the images Frisco (**a**) and Diego (**b**), $\sigma^2 = 100$.

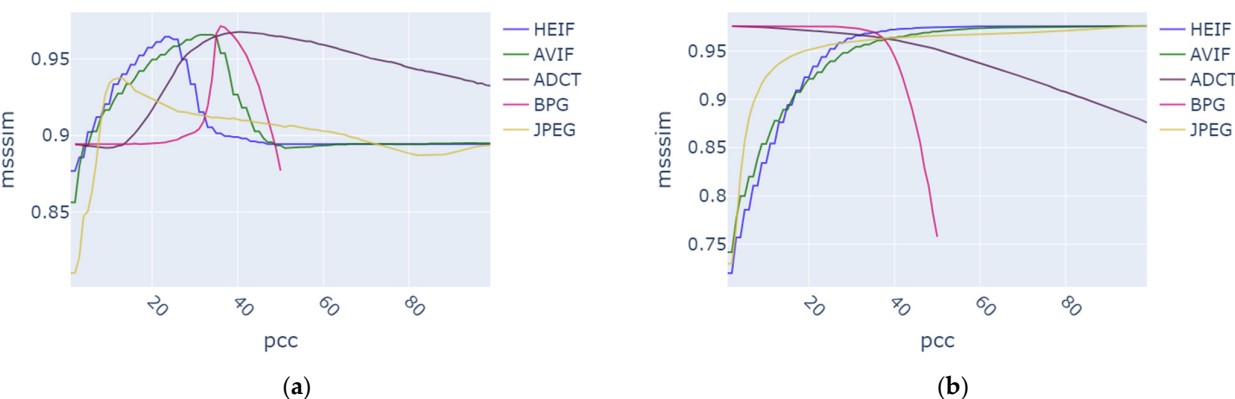

**Figure 7.** Dependencies MS-SSIM$_{tc}$(PCC) for the images Frisco (**a**) and Diego (**b**), $\sigma^2 = 100$.

**Figure 8.** Dependencies PSNR$_{tc}$(PCC) for the images Frisco (**a**) and Diego (**b**), $\sigma^2 = 25$.

The dependencies PSNR$_{tc}$(PCC) for $\sigma^2 = 196$ are given in Figure 9. OOPs are observed for all five encoders for the image Frisco where the best PSNR$_{OOP}$ is seen for the BPG encoder and the worst for JPEG (Figure 9a). It is interesting that OOPs appear for the test image Diego (Figure 9b) for three out of five coders. Although formally for AVIF and JPEG OOPs are absent, the corresponding dependencies have local maxima. Joint analysis of the dependencies in Figures 9 and 10 shows the following. First, again, OOPs for the

considered metrics PSNR and PSNR-HVS-M are observed for the same PCC values (for a given image, compare data in Figures 9a and 10a). Second, it might be that an OOP exists according to one metric (PSNR, Figure 9b) but does not exist according to another metric (PSNR-HVS-M, Figure 10b). We have observed this phenomenon earlier [18,37] that OOPs for visual quality metrics are observed more rarely than for PSNR.

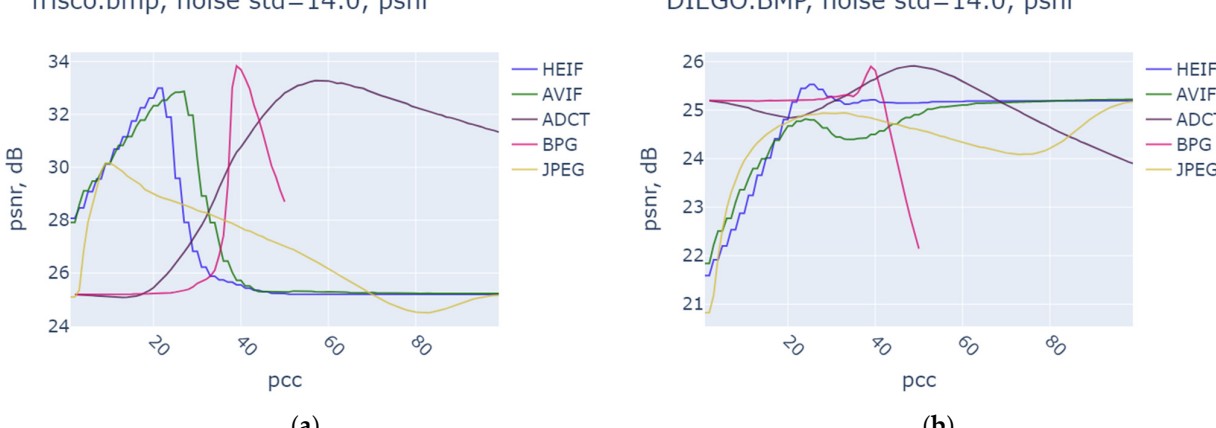

**Figure 9.** Dependencies $PSNR_{tc}$(PCC) for the images Frisco (**a**) and Diego (**b**), $\sigma^2 = 196$.

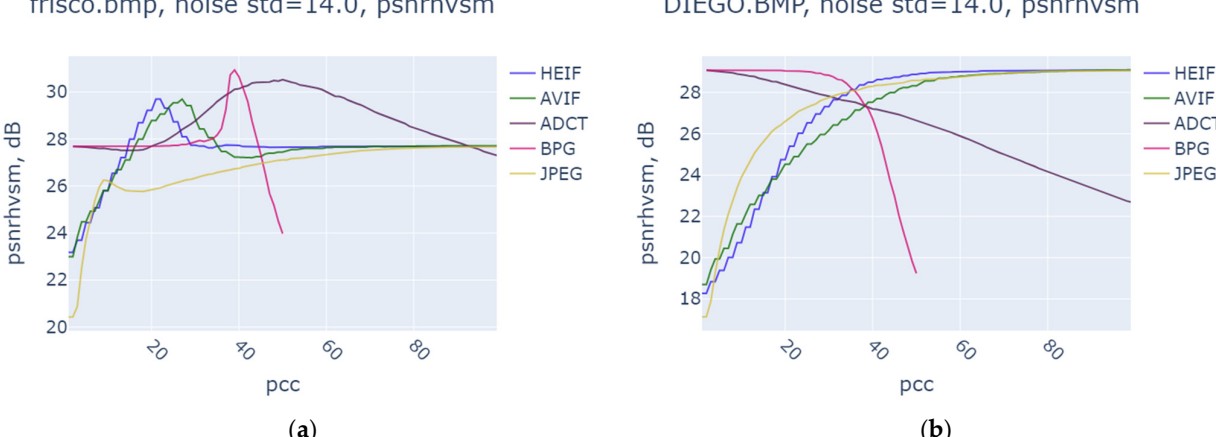

**Figure 10.** Dependencies $PSNR\text{-}HVS\text{-}M_{tc}$(PCC) for the images Frisco (**a**) and Diego (**b**), $\sigma^2 = 196$.

The dependencies of metrics on the PCC do not allow us to compare the coders' performance clearly. Because of this, we have calculated the dependencies $PSNR_{tc}$(CR)—they are presented in Figure 11 for $\sigma^2 = 100$. As one can see, for the image Frisco, maximal values of $PSNR_{tc}$ are observed for different values of CR for different coders: $CR_{OOP}$ is approximately equal to 30 for JPEG, 40 for ADCT, 42 for BPG, 46 for AVIF, and 48 for HEIF coders, respectively. Then, it might seem that HEIF and AVIF are better than the BPG and ADCT coders. However, an analysis of the curves in Figure 11a clearly shows that in the area of CR under interest (from 30 to 70), the largest $PSNR_{tc}$ values are provided by the BPG encoder. The analysis for the test image Diego (Figure 11b) shows that there are no OOPs (this is in agreement with data in Figure 5b) and the best results (the largest $PSNR_{tc}$) are again provided by the BPG encoder in a wide range of CR values. For the other three encoders, the difference is negligible starting from CR = 20. JPEG obviously produces the worst results.

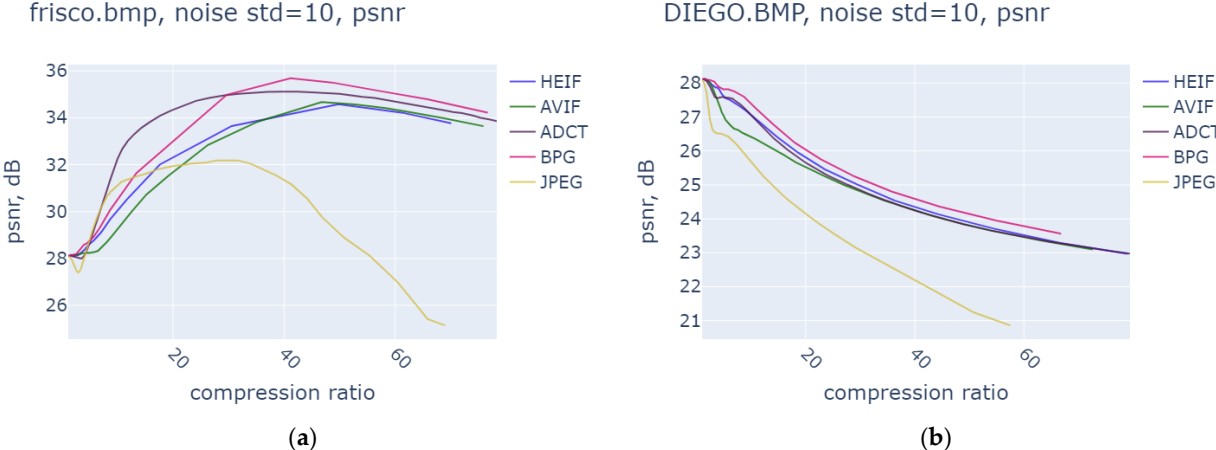

**Figure 11.** Dependencies PSNR$_{tc}$(CR) for the images Frisco (**a**) and Diego (**b**), $\sigma^2 = 100$.

It is worth comparing the coders' performance for other noise-intensity and quality metric. Figure 12 presents such data for the test image Diego corrupted by AWGN with a variance of 196. According to the PSNR$_{tc}$ (Figure 12a), an OOP exists for three coders (and local maxima exist for two other coders) where the BPG encoder is again the best. According to MS-SSIM$_{tc}$ (Figure 12b) and PSNE-HVS-M$_{tc}$ (Figure 12c), an OOP does not exist, but the BPG encoder performs better than the others.

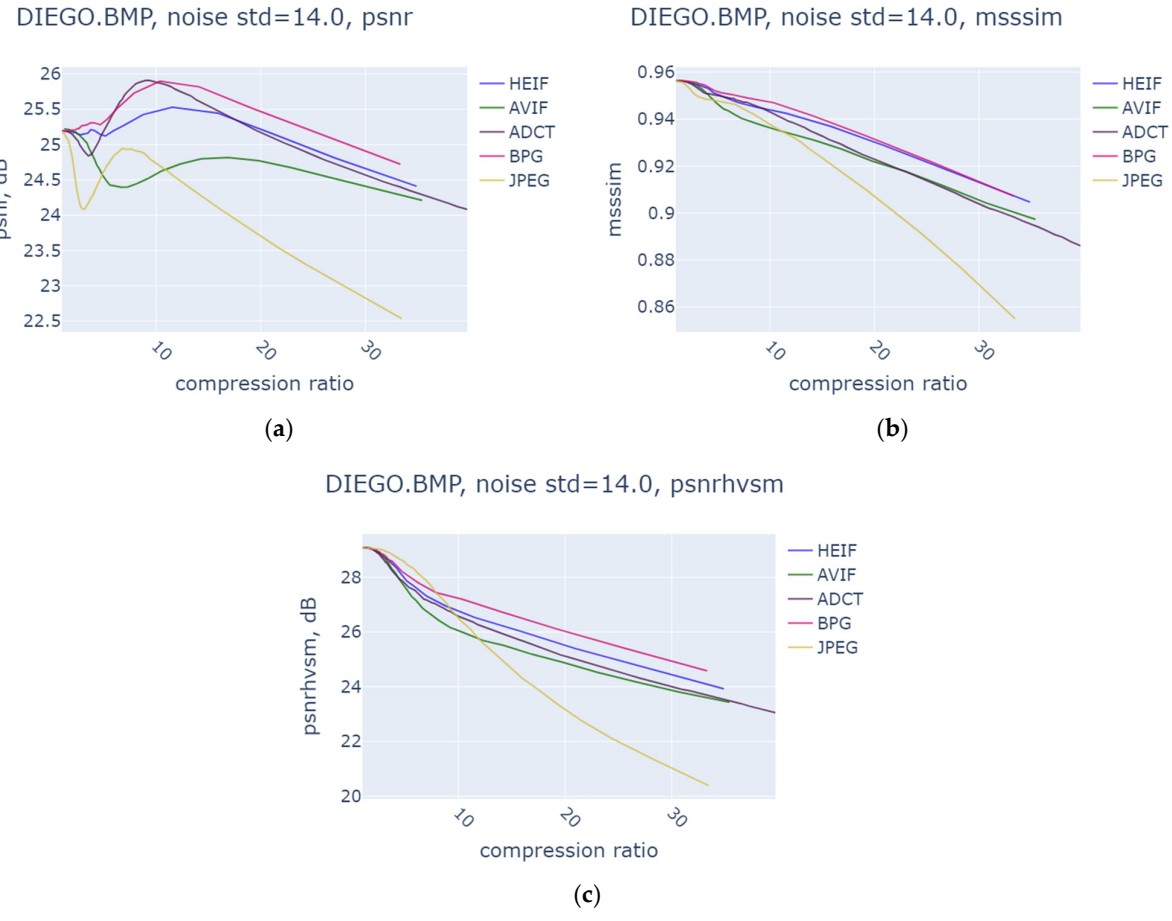

**Figure 12.** Dependencies PSNR$_{tc}$(CR) (**a**), MS-SSIM$_{tc}$(CR) (**b**), and PSNR-HVS-M$_{tc}$(CR) (**c**) for the image Diego (**b**), $\sigma^2 = 196$.

## 4. Discussion

It could be nice if the PCC to reach the OOP for a given coder and noise variance was set automatically under the assumption that the noise variance is a priori known or accurately estimated in a blind manner in advance. Such opportunities have been already demonstrated for the AGU, ADCT, and BPG encoders.

From previous experience [32–34,37], it can be expected that the OOP for AVIF and HEIF depends on the image, metric, coder, and noise intensity. To establish all dependencies, a thorough study is needed. Because of this, let us try to obtain the main tendencies. For this purpose, some data are collected in Tables 1–3 for situations when an OOP is observed. We do not want to study the main tendencies for the BPG encoder since they have been already investigated in [16]. The data are mainly presented for comparison purposes.

**Table 1.** Compression characteristics in OOP according to PSNR for Frisco.

| Noise Variance | JPEG | | AVIF | | HEIF | | BPG | |
|---|---|---|---|---|---|---|---|---|
| | $PSNR_{max}$ | CR | $PSNR_{max}$ | CR | $PSNR_{max}$ | CR | $PSNR_{max}$ | CR |
| 25 | 36.5 (QF = 48) | 13.2 | 38.0 (QF = 51) | 26.5 | 37.7 (QF = 33) | 31.6 | 39.3 (Q = 30) | 22.4 |
| 100 | 32.2 (QF = 14) | 29.8 | 34.7 (QF = 35) | 46.8 | 34.6 (QF = 25) | 49.8 | 35.7 (Q = 36) | 41.2 |
| 196 | 30.2 (QF = 10) | 34.7 | 32.9 (QF = 27) | 61.2 | 33.0 (QF = 21) | 66.0 | 33.8 (Q = 39) | 56.6 |

**Table 2.** Compression characteristics in OOP according to MS-SSIM for Frisco.

| Noise Variance | JPEG | | AVIF | | HEIF | | BPG | |
|---|---|---|---|---|---|---|---|---|
| | $MS\text{-}SSIM_{max}$ | CR | $MS\text{-}SSIM_{max}$ | CR | $MS\text{-}SSIM_{max}$ | CR | $MS\text{-}SSIM_{max}$ | CR |
| 25 | 0.9743 (QF = 33) | 18.3 | 0.9814 (QF = 49) | 30.2 | 0.9815 (QF = 33) | 31.6 | 0.9856 (Q = 30) | 22.4 |
| 100 | 0.9738 (QF = 13) | 31.7 | 0.9659 (QF = 33) | 57.8 | 0.9645 (QF = 23) | 61.1 | 0.9716 (Q = 36) | 41.2 |
| 196 | 0.9002 (QF = 7) | 45.3 | 0.9526 (QF = 25) | 69.3 | 0.9508 (QF = 21) | 66.0 | 0.9601 (Q = 39) | 56.6 |

**Table 3.** Compression characteristics in OOP according to PSNR for Diego.

| Noise Variance | JPEG | | AVIF | | HEIF | | BPG | |
|---|---|---|---|---|---|---|---|---|
| | $PSNR_{max}$ | CR | $PSNR_{max}$ | CR | $PSNR_{max}$ | CR | $PSNR_{max}$ | CR |
| 196 | 24.9 (QF = 28) | 7.6 | 24.8 (QF = 24 *) | 16.9 | 25.5 (QF = 25) | 11.5 | 25.9 (Q = 39) | 10.4 |

\* Data for local maximum.

For AVIF (Table 1), an increase in noise variance leads to a smaller maximal $PSNR_{tc}$ (denoted as $PSNR_{max}$), smaller QF, and larger CR. The same holds for HEIF but there are some differences. The $PSNR_{max}$ values are practically the same, the $QF_{OOP}$ values are smaller, and the $CR_{OOP}$ values are slightly larger than for AVIF.

According to the MS-SSIM (Table 2), an increase in noise intensity results in a smaller maximal $MS\text{-}SSIM_{tc}$ ($MS\text{-}SSIM_{max}$), smaller QF, and larger CR for AVIF. The same tendencies hold for HEIF. In general, the optimal QF values according to MS-SSIM are slightly smaller than according to PSNR, but the difference is not essential (compare the corresponding dependencies in Figures 5a and 7a).

For the test image Diego, there are only data for $\sigma^2 = 196$ (Table 3). Comparing the optimal values of QF for the images Frisco and Diego for $\sigma^2 = 196$ (Tables 1 and 3), it is seen that they are quite close but not the same.

To show the direction of further research, Figure 13 presents one more test image (Fr02) and the plots for it ($\sigma^2 = 100$). This image is more complex than Frisco but less complex than Diego. As seen, OOPs exist for four coders and the best results are again observed for the BPG encoder. The OOPs are less obvious ($PSNR_{max}$ values are smaller) than for the image Frisco (Figure 11a), and the CR values in the OOP are also significantly smaller. The QF in the OOP is equal to 35 for AVIF and 31 for HEIF. Thus, the values of $QF_{OOP}$ are slightly other than in the previous considered case (Table 1), at least, for the HEIF encoder. Because of this, we analyzed several other test images. The obtained results are the following. For $\sigma^2 = 100$, $QF_{OOP}$ for AVIF is within the limits of 32–35 and for HEIF the limits are from 23 to 31. For $\sigma^2 = 196$, $QF_{OOP}$ varies within the limits of 22–27 for AVIF and from 20 to 28 for HEIF. Thus, the limits are quite wide and there is a tendency for a reduction in $QF_{OOP}$ if $\sigma^2$ increases. Maybe the position of OOP depends not only on the noise intensity but on the image complexity. This hypothesis needs verification in the future. Maybe then, non-iterative procedures for determining $QF_{OOP}$ will be proposed.

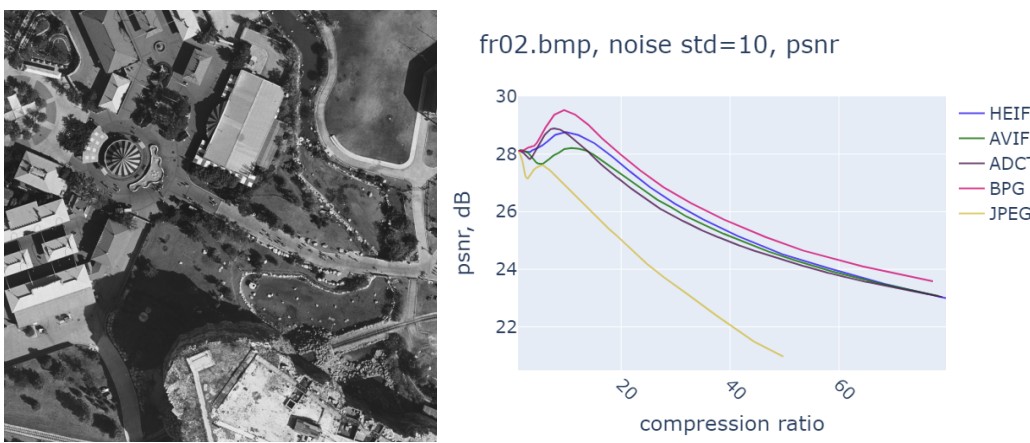

**Figure 13.** The test image Fr02 and dependencies $PSNR_{tc}(CR)$ for it, $\sigma^2 = 100$.

Meanwhile, currently there exists another opportunity to reach the OOP. As mentioned above, the following fact [32] was observed for the OOP for JPEG, JPE2000, and some other coders: $MSE_{nc}$ was approximately equal to $\sigma^2$. Let us check whether or not this happens for the AVIF and HEIF encoders. Figure 14 presents some dependencies of the $MSE_{nc}$ on the PCC. In Figure 14a (image Frisco, $\sigma^2 = 25$), $MSE_{nc} = 25$ is seen for $QF \approx 36$ for HEIF and for $QF \approx 50$ for AVIF. This is in good agreement with the data in Table 1. In Figure 14b (image Frisco, $\sigma^2 = 100$), $MSE_{nc} = 100$ is observed for $QF \approx 26$ for HEIF and for $QF \approx 37$ for AVIF. These data are in agreement with the corresponding data in Table 1 for $\sigma^2 = 100$. In Figure 14c (image Frisco, $\sigma^2 = 196$), $MSE_{nc} = 196$ is seen for $QF \approx 22$ for HEIF and for $QF \approx 27$ for AVIF. Again, agreement with the corresponding data in Table 1 for $\sigma^2 = 196$ is observed. Finally, Figure 14b presents the plots for the image Diego, $\sigma^2 = 196$. $MSE_{nc} \approx \sigma^2$ is observed for $QF \approx 24$ for HEIF and for $QF \approx 27$ for AVIF which agrees with the corresponding data in Table 3.

Thus, the property that $MSE_{nc} \approx \sigma^2$ in the OOP is observed for AVIF and HEIF as well for other encoders [32]. It allows us to propose quite a simple algorithm (further called Algorithm 1, see below) for reaching the OOP:

(1) Suppose that $\sigma^2$ is known in advance; if it is unknown, estimate it;
(2) Set a starting $QF_{st}$ for a used coder according to the observations given above (see also the data in Table 4);
(3) Compress and decompress a considered image using $QF_{st}$ and calculate $MSE_{nc}$;
(4) If $0.9\sigma^2 \leq MSE_{nc} \leq 1.1\sigma^2$, retain the compressed image obtained at Step 3 as the final one; if $MSE_{nc} \leq 0.9\sigma^2$, decrease QF by 2 and continue; if $MSE_{nc} > 1.1\sigma^2$, increase QF by 2 and continue;

(5) For the new QF, compress and decompress the image, calculate $MSE_{nc}$, and continue checking the validity of $0.9\sigma^2 \le MSE_{nc} \le 1.1\sigma^2$ as in Step 4; stop when it is valid and retain the last obtained compressed image as the final one.

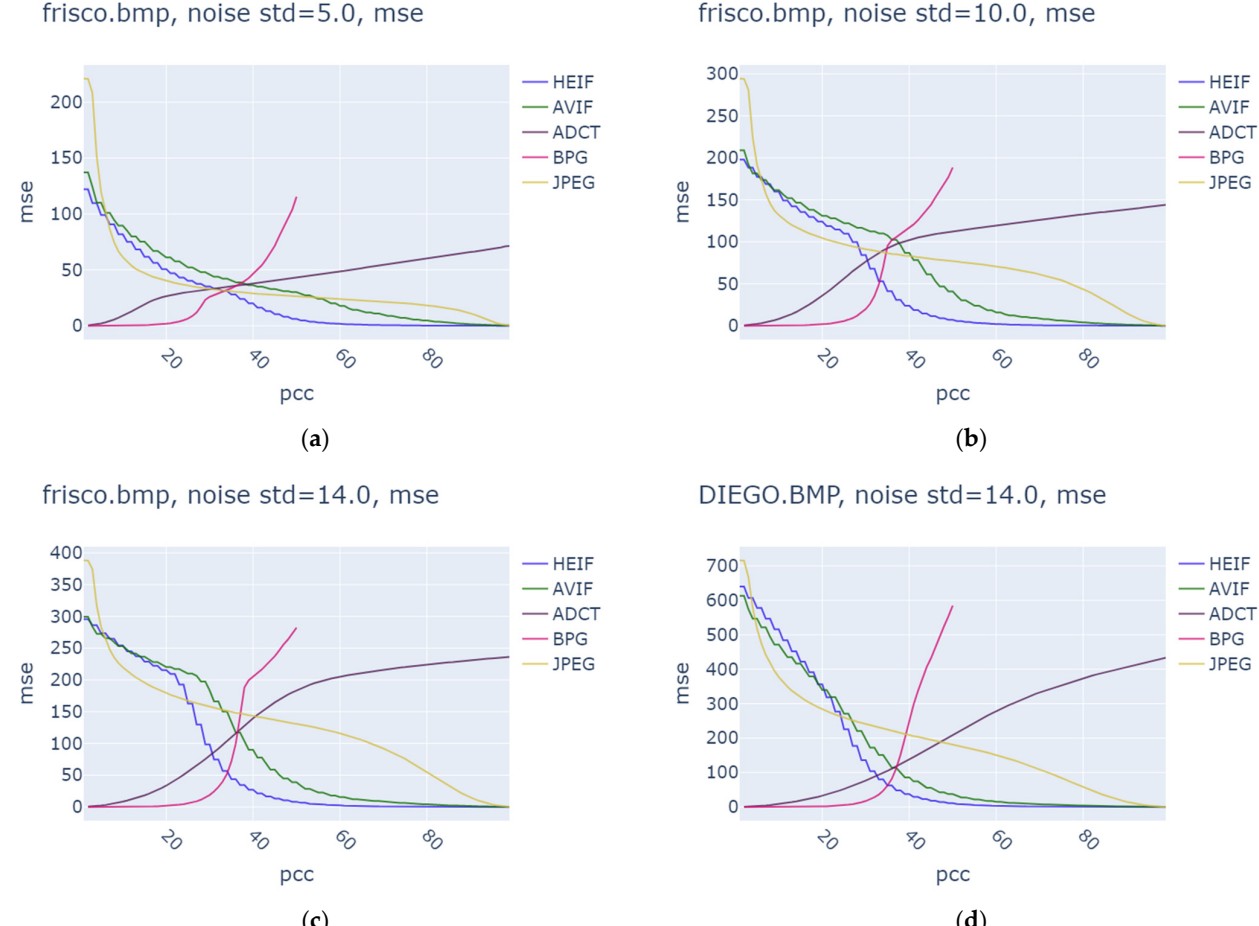

**Figure 14.** Dependencies of $MSE_{nc}$ on PCC for Frisco (**a**–**c**) and Diego (**d**).

**Table 4.** Recommended $QF_{st}$ for AVIF and HEIF.

| $\sigma^2$ | 20–30 | 31–44 | 45–64 | 65–90 | 91–130 | 131–180 | >180 |
|---|---|---|---|---|---|---|---|
| **$QF_{st}$ for AVIF** | 50 | 45 | 41 | 37 | 33 | 30 | 25 |
| **$QF_{st}$ for HEIF** | 33 | 31 | 29 | 27 | 26 | 25 | 24 |

Note that we do not give any recommendation for $\sigma^2 < 20$ since the noise in such images is invisible. The recommendations in Table 4 are based on the materials and data presented above as well on data obtained for intermediate values of $\sigma^2$. The recommendations given in Table 4 allow us to minimize (on the average) the number of iterations needed to reach the OOP.

As one can see, the noise variance $\sigma^2$ is used in the proposed algorithm as the parameter. Although this algorithm produces $MSE_{nc} \approx \sigma^2$ and it is unable to provide $MSE_{nc} = \sigma^2$ since QF can only be an integer, errors of the noise variance estimation can produce additional errors in the determination of the optimal QF. The negative impact of such errors is negligible if the relative error of noise variance estimation is less than 5%—see the examples of $MSE_{nc}$ changing with QF changing by 2 below.

We have checked the algorithm proposed above for the image Fr02 (Figure 13a) corrupted by AWGN with a variance equal to 100.

For AVIF, according to recommendations in Table 4, we use $QF_{st} = 33$ and obtain $MSE_{nc} = 131.7$ ($MSE_{tc} = 98.7$). According to the proposed algorithm, we have to increase QF by 2. For QF = 35, one obtains $MSE_{nc} = 121.1$ ($MSE_{tc} = 98.1$), i.e., closer to the desired value. At the next step, for QF = 37, $MSE_{nc} = 108.8$ ($MSE_{tc} = 99.0$), i.e., the algorithm can be stopped. In fact, QF = 35 corresponds to the OOP but the final result, QF = 37, produces a result that is only slightly worse.

For HEIF, according to the data in Table 4, we have to start from $QF_{st} = 26$ and then obtain $MSE_{nc} = 169.0$ ($MSE_{tc} = 104.8$). Then, QS has to be increased by 2 and, for QF = 28, one obtains $MSE_{nc} = 148.8$ ($MSE_{tc} = 94.5$). At the next step, for QS = 30, $MSE_{nc} = 128.1$ ($MSE_{tc} = 88.5$). Finally, for QF = 32, one has $MSE_{nc} = 107.8$ ($MSE_{tc} = 86.4$). The algorithm stops and $MSE_{tc} = 86.4$ corresponds to the OOP. Thus, in this case, we have exactly reached the OOP.

As one can see, three and four iteration steps were enough to reach the OOP or its close neighborhood. Probably, some other algorithms of reaching the OOP are possible.

Let us also present some results of noisy image compression in the OOP. Figure 15a shows the noisy image Frisco, $\sigma^2 = 100$. The image compressed by AVIF with QF = 36 is presented in Figure 15b whilst the same image compressed by HEIF with QF = 25 is shown in Figure 15c. Finally, the image compressed by BPG with Q = 35 is demonstrated in Figure 15d. As seen, the noise in all of the compressed images is significantly suppressed whilst the details are preserved well enough. We did not notice any sufficient differences between images compressed by the AVIF, HEIF, and BPG encoders in the corresponding optimal operation points (Figure 15b–d).

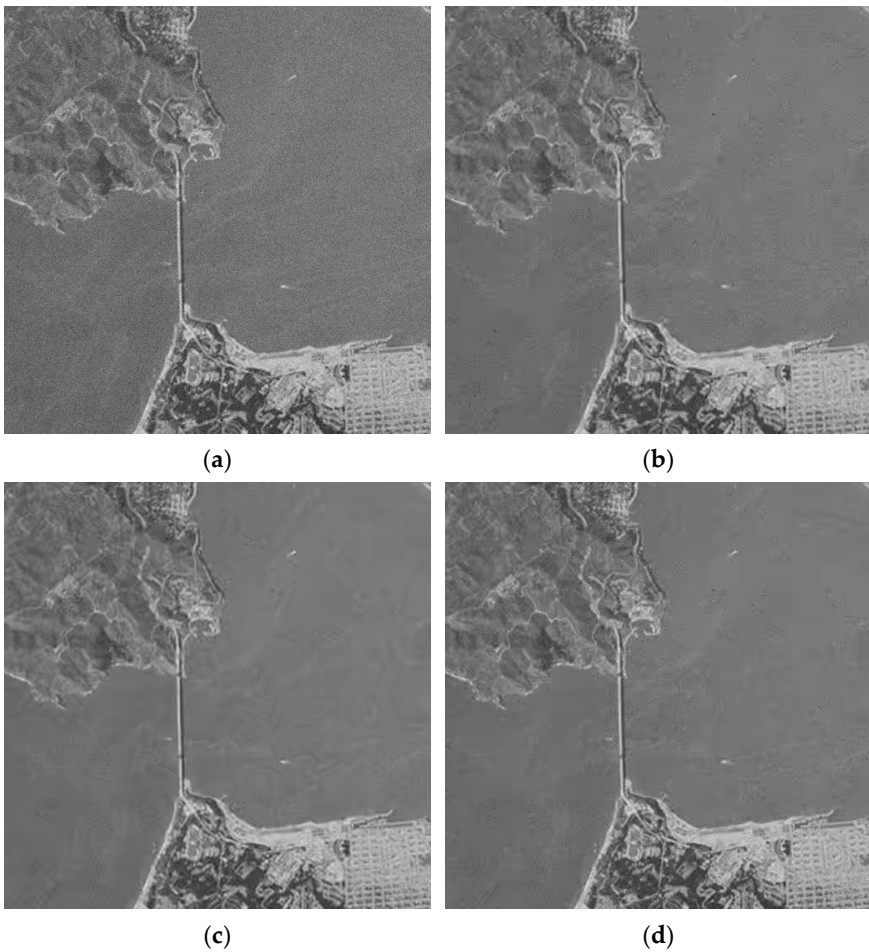

(**a**)  (**b**)

(**c**)  (**d**)

**Figure 15.** Noisy image Frisco (**a**) and results of its compression in the OOP by AVIF (**b**), HEIF (**c**), and BPG (**d**).

All studies above are presented for AWGN which, as already mentioned, is the simplified model. Hence, let us consider a practical case when noise is signal-dependent as it usually happens to be for hyperspectral data. As an example, below we consider Hyperion data, namely a fragment of the image E01H1800252002116110KZ, the 15th sub-band, presented in Figure 16a. The input PSNR for this image is smaller than 40 dB [11]: the minimal and maximal values are equal to 0 and 3000, respectively; the noise variance model is

$$\sigma_{ij}^2 = \sigma_0^2 + kI_{ij}^{true} \tag{3}$$

where $\sigma_0^2$ is variance of the signal-independent (additive) component and k denotes the parameter of the signal-dependent noise component where the estimated values of these parameters are equal to 564 and 0.271, respectively. The equivalent noise variance equals to 1165 and this means that the contributions of the signal-independent and signal-dependent noise components are approximately the same. Noise can be noticed in bright regions of the sub-band image.

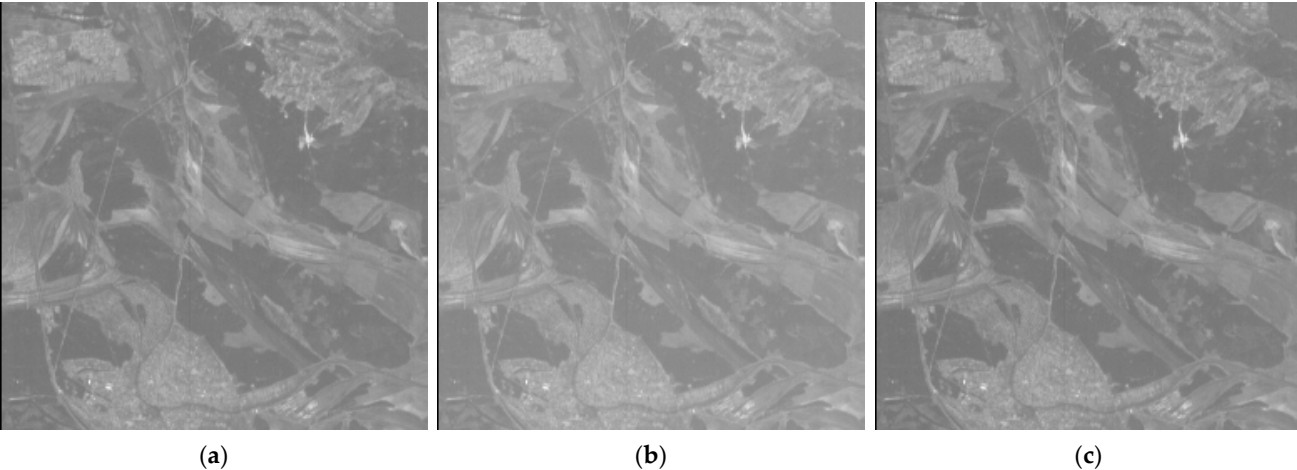

|  (**a**)  |  (**b**)  |  (**c**)  |

**Figure 16.** Visualized original image fragment (**a**), the corresponding image $I^{GA1}$ (**b**), and the image after inverse transform (without compression) (**c**).

Since we have the algorithm suited for additive noise, let us use a variance-stabilizing transform adapted to the aforementioned model of signal-dependent noise. This is the so-called generalized Anscombe transform expressed as

$$I_{ij}^{GA} = \left(\frac{2}{k}\right)\sqrt{kI_{ij}^n + \frac{3}{8k^2} + \sigma_0^2} \tag{4}$$

Its use for the data we analyze leads to the unity variance of the noise. For our case, the range of the $I_{ij}^{GA}$ values was from 175 to 290. To make the data fit the standard range from 0 to 255, we have applied the transform $I_{ij}^{GA1} = 2.22(I_{ij}^{GA} - 175)$ and, thus, obtained a noise variance slightly smaller than 5. The image $I^{GA1}$ is shown in Figure 16b. Note that the distortions introduced by the direct transform and the corresponding inverse transform are characterized by an $MSE_{dit}$ equal to 22.0, i.e., these distortions are considerably smaller than the equivalent noise variance and can be neglected (see the image in Figure 16c that seems identical to the image in Figure 16a).

Lossy compression providing $MSE_{nc} \approx 5$ was carried out for the image $I^{GA1}$. The following values were obtained for different coders. For BPG, Q = 24 and CR = 8.3 (for the original image, $MSE_{nc} \approx 1036.3$). For HEIF, QF = 50 and CR = 8.13 (for the original image, $MSE_{nc} \approx 1100.9$). Finally, for AVIF, QF = 76 and CR = 7.38 (for the original image, $MSE_{nc} \approx 1064.5$). Thus, we have provided an $MSE_{nc}$ for the original image approximately equal to the equivalent noise variance (1165). Note that the CR values are given with

respect to 8-bit data. Being calculated with respect to the original 16-bit data, the CR values are twice as large.

The compressed images (after inverse VST) are shown in Figure 17 for all three considered coders. As one can see, they are all very similar to the original image in Figure 16a.

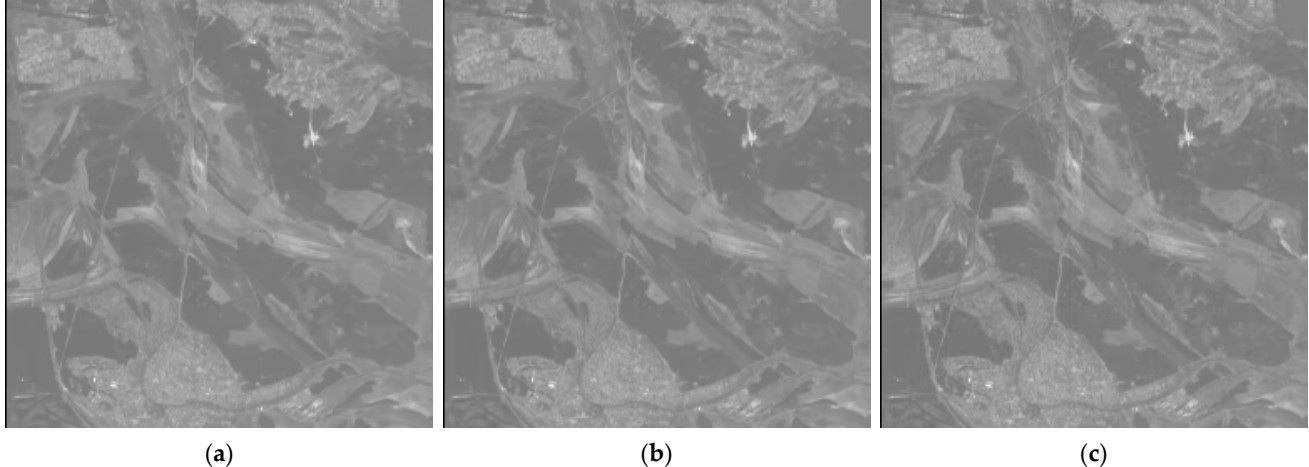

(**a**)          (**b**)          (**c**)

**Figure 17.** Visualized compressed images for the BPG (**a**), HEIF (**b**), and AVIF (**c**) coders (after inverse transform).

Above, we have concentrated on the case of grayscale (single-channel) images contaminated by AWGN (Sections 2 and 3) and the case of signal-dependent noise (Section 4) represented by 8-bit integers (Sections 2 and 3) or normalized to the 0–255 range (Section 4). For more general cases of signal-dependent noise and other ranges of acquired data representation, the automatic compression procedure is shown in Figure 18. Note that not all the blocks in Figure 18 may be needed. Block 1 may not be used if the noise characteristics are known in advance. Block 2 is not needed if the noise is purely additive. Block 3 is not needed if the acquired image is in the range of 0–255.

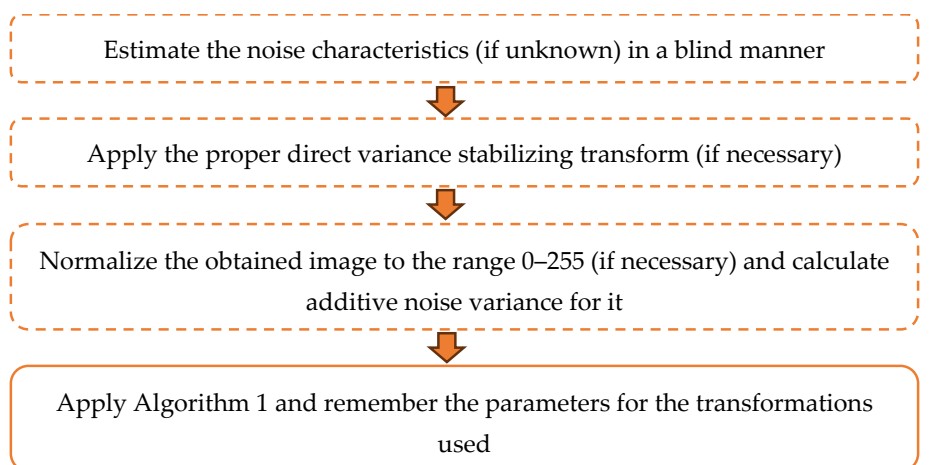

**Figure 18.** Flow-chart of image compression procedure.

Decompression should be carried out in inverse order. We do not concentrate on the variance-stabilizing transforms to be used, assuming that this aspect is out of the paper's scope (for pure multiplicative noise, the logarithmic-type transform can be applied [55]).

If a multichannel image has to be compressed, one has to know the noise properties (at least, the input $PSNR^n$ in component images). If the $PSNR^n$ is relatively high (e.g., larger than 35 dB), the noise presence can be neglected and lossy compression of the corresponding component images can be performed with quality (distortion) control (see, e.g., [15]). If the

$PSNR^n$ is relatively low (e.g., does not exceed dB), lossy compression in the OOP can be applied as described above. This can be performed component-wise. Meanwhile, we expect that better compression can be provided by the joint (three-dimensional) compression of several noisy component images [54]. This can be a direction of future studies.

Finally, let us present some data about computational expenses. We used a Python port of the libheif library (https://github.com/strukturag/libheif, accessed on 1 March 2024) as the HEIF/AVIF codec. As it is claimed in the library documentation, libheif is an ISO/IEC 23008-12:2017 [56] HEIF and AVIF (AV1 Image File Format) file format decoder and encoder. There is partial support for ISO/IEC 23008-12:2022 [56] (2nd Edition) capabilities; libheif makes use of libde265 for HEIF image decoding and x265 for encoding. For AVIF, libaom, dav1d, svt-av1, or rav1e are used as codecs. For the BPG encoder, we used the source available at https://bellard.org/bpg/ (accessed on 1 March 2024) [17].

We used a laptop equipped with Intel(R) Core(TM) i7-4710HQ CPU @ 2.50GHz, 16 GB RAM, Windows 10 x64. The grayscale image size was $512 \times 512$ pixels. We analyzed two values of PCC for each coder. One of them (Q = 2 for BPG and QF = 98 for AVIF and HEIF) relates to a small CR; the other one (Q = 40 for BPG and QF = 30 for AVIF and HEIF) corresponds to the main range of coder operation. Since the compression and decompression time differ significantly, we give them separately. The obtained data are presented in Table 5.

**Table 5.** Computation expenses for compression and decompression.

| Image | Encoder | PCC | Compression Time | Decompression Time |
|---|---|---|---|---|
| Frisco.bmp | BPG | 2 | 0.69 | 0.19 |
| Frisco.bmp | BPG | 40 | 0.24 | 0.12 |
| DIEGO.BMP | BPG | 2 | 0.74 | 0.24 |
| DIEGO.BMP | BPG | 40 | 0.36 | 0.19 |
| fr02.bmp | BPG | 2 | 0.73 | 0.22 |
| fr02.bmp | BPG | 40 | 0.34 | 0.18 |
| frisco.bmp | HEIF | 98 | 0.37 | 0.04 |
| frisco.bmp | HEIF | 30 | 0.15 | 0.02 |
| DIEGO.BMP | HEIF | 98 | 0.4 | 0.06 |
| DIEGO.BMP | HEIF | 30 | 0.23 | 0.04 |
| fr02.bmp | HEIF | 98 | 0.44 | 0.06 |
| fr02.bmp | HEIF | 30 | 0.2 | 0.04 |
| frisco.bmp | AVIF | 98 | 0.23 | 0.05 |
| frisco.bmp | AVIF | 30 | 0.09 | 0.02 |
| DIEGO.BMP | AVIF | 98 | 0.34 | 0.07 |
| DIEGO.BMP | AVIF | 30 | 0.17 | 0.03 |
| fr02.bmp | AVIF | 98 | 0.31 | 0.06 |
| fr02.bmp | AVIF | 30 | 0.21 | 0.03 |

Their analysis shows the following. First, decompression is significantly faster than compression. Second, the simple-structure image (Frisco) is compressed and decompressed faster than complex-structure ones (Diego and Fr02) for all considered encoders. Third, AVIF and HEIF seem to be faster than BPG (although it might also depend on software or hardware implementation).

Meanwhile, even in the worst case of complex images and a small CR, the computation time is rather small (recall that all considered encoders are used in video coding, this shows that they are fast enough). Compression can be accelerated using FPGA [57].

## 5. Conclusions

The paper deals with the lossy compression of single-channel noisy images. The main emphasis has been paid to the AVIF and HEIF coders and a comparison of their performance to some known counterparts. We have demonstrated that, similarly to other modern coders, AVIF and HEIF might have an OOP where the OOP might exist according

to different quality metrics. Comparison has shown that compressed image characteristics (PSNR and visual quality metrics) in the OOP for AVIF and HEIF are slightly worse than for the BPG encoder, but the CR in the OOP is slightly larger. If the noise intensity increases, the optimal QF for AVIF and HEIF decreases. The ranges of the optimal QF have been determined. A procedure (algorithm) for reaching the OOP has been proposed. Examples for real-life data corrupted by signal-dependent noise are presented.

We have not considered the task of predicting whether or not the OOP exists for AVIF or HEIF. However, taking into account that such a task has been already solved for the BPG and ADCT encoders, it seems possible that it will be also solved for AVIF and HEIF. It is also worth considering in the future the cases of signal-dependent noise and multichannel RS images.

**Author Contributions:** Conceptualization, V.L. and B.V.; methodology, V.L.; software, S.K.; validation, S.K. and B.V.; formal analysis, S.K. and V.L.; investigation, S.K.; writing—original draft preparation, V.L.; writing—review and editing, B.V.; visualization, S.K. and B.V.; supervision, V.L. and B.V. All authors have read and agreed to the published version of the manuscript.

**Funding:** The research performed in this manuscript was partially supported by the French Ministries of Europe and Foreign Affairs (MEAE) and Higher Education, Research and Innovation (MESRI) through the PHC Dnipro 2021 Project No. 46844Z.

**Data Availability Statement:** Data are contained within the article.

**Conflicts of Interest:** The authors declare no conflicts of interest.

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
