# Peer review of "Lossy Compression of Single-channel Noisy Images by Modern Coders"

_remotesensing, doi:10.3390/rs16122093_

Round 1

Reviewer 1 Report

Comments and Suggestions for Authors

This manuscript mainly discusses the problem of lossy compression of remote sensing images, especially the compression effect when there is noise in the image. The paper analyzes the performance of several modern encoders ( such as AVIF and HEIF ) in compressing noisy images, and compares them with JPEG, JPEG2000 and encoders based on discrete cosine transform ( DCT ). It is found that the existence of OOP depends on the complexity of the image and the noise intensity. The manuscript is generally well-structured, and the contributions are clearly articulated. However, there are areas that require further clarification and expansion to strengthen the manuscript.
1. The time complexity of the proposed method should be given and compared with the latest technology in the field of image compression.
2. The author only discusses the additive white Gaussian noise, so whether it has a similar effect on other types of noise, please simply analyze some of the noise types it applies to.
3. In practical applications, it is very difficult to accurately estimate the noise variance. If the noise estimation is not accurate, how much impact will it have on the determination of OOP ?
4. The paper mainly focuses on single-channel images, but in remote sensing images, multi-channels are common. Is the proposed method suitable for multi-channel images ?

Author Response

First of all, we would like to thank anonymous reviewer for valuable comments and propositions. Our replies are on item-to-item basis and in Italic (all the changes in the revised version are marked by red color).

  1. The time complexity of the proposed method should be given and compared with the latest technology in the field of image compression.

Thanks for this comment. AVIF and HEIF time complexity is now compared to the BPG encoder time complexity at the very end of Section 4 in the lower half of page 16 and in Table 5. AVIF and HEIF time complexity is shown preferable. 

  1. The author only discusses the additive white Gaussian noise, so whether it has a similar effect on other types of noise, please simply analyze some of the noise types it applies to.

Thanks for this comment. In fact, we discuss what to do if noise is signal-dependent and, moreover, give an example in Section 4. We first mention signal-dependent noise in page 2 and give references to the papers [27, 42]. Signal-dependent noise is also mentioned in page 3 and the possibility of using a proper variance stabilizing transform is stressed [42, 55]. After this, we consider a practical case of signal-dependent noise in Section 4 (pages 14 and 15 of the revised paper). We have also added the flow-chart of image compression procedure and its description in pages 13 and 16.   

  1. In practical applications, it is very difficult to accurately estimate the noise variance. If the noise estimation is not accurate, how much impact will it have on the determination of OOP?

Two authors of this paper (B. Vozel and V. Lukin) dealt with blind estimation of noise characteristics a lot and fully agree with reviewer’s comment. Inaccurate estimation of noise variance might have negative impact on OOP determination, but this effect is negligible if the relative error of noise variance estimation is less than 5% - see the examples of MSEnc changing with QF changing by 2 in Section 4. We have added brief explanation in page 13 of the revised version. Probably, the task is worth additional consideration. Meanwhile, we can state that blind estimation of noise characteristics with aforementioned accuracy is often possible (although not always).    

  1. The paper mainly focuses on single-channel images, but in remote sensing images, multi-channels are common. Is the proposed method suitable for multi-channel images ?

It is, of course, suitable for component-wise compression of multichannel images. We are pretty sure that the method can work in 3D manner with higher efficiency compared to component-wise compression but this has to be additionally studied. We briefly consider this aspect in page 16 of the revised paper and point its importance in Conclusions. Some aspects (for other coders) have been earlier considered in our paper [54].

Reviewer 2 Report

Comments and Suggestions for Authors

The study's focus on the performance of modern coders like AVIF and HEIF in the presence of noise is commendable. The impact of noise on lossy compression of remote sensing images is analyzed. The study is well-prepared, designed, and conducted. The research is meaningful and innovative. Other comments include:

1. The writing style is succinct, though at points it may be too technical for readers not familiar with the field. Consider clarifying technical terms or abbreviations when they first appear in the text. In the introduction, it would be helpful to briefly describe why understanding the effects of noise is important for practical applications.

2. The manuscript studies the analysis of optimal operation points on AVIF and HEIF encoder compression for remotely sensed images with additive white Gaussian noise. The manuscript has some limitations, e.g., only single-channel images are studied for remotely sensed images, only additive Gaussian white noise is studied for noisy images, and five editors are studied for compression. Is it still generalizable to other forms of remotely sensed imagery, other forms of noisy imagery, and other forms of compression?

3. What is the specific information about the two remote sensing images involved in the experiment, Frisco and Diego? Are they panchromatic images or single-channel images obtained after multispectral image processing or others? What are their sources?

4. The description of the techniques associated with the experiments and the analysis of the experiments are mixed, which greatly reduces the readability of the manuscript.

5. An overall flow diagram is required to enhance understandability.

6. Too many outdated references are cited and it is recommended to look for more recent and cutting-edge references.

7. The analysis for the additive white Gaussian noise is a good choice for the study due to its commonality in practical situations. However, the manuscript should mention if other types of noise will also be considered or if there are plans for future work in this area.

Comments on the Quality of English Language

Minor editing of English language required.

Author Response

First of all, we would like to thank anonymous reviewer for valuable comments and propositions. Our replies are on item-to-item basis and in Italic (all the changes in the revised version are marked by red color).

  1. The writing style is succinct, though at points it may be too technical for readers not familiar with the field. Consider clarifying technical terms or abbreviations when they first appear in the text. In the introduction, it would be helpful to briefly describe why understanding the effects of noise is important for practical applications.

Thanks for this comment. We have tried to give more explanations and illustrations in the revised version. We have also checked abbreviations. Please do not ask about the abbreviation AGU J This coder was mainly designed by our colleague Ponomarenko who just got his son at that period of time and AGU is of the first words uttered. We have also added Fig. 1 demonstrating the examples of noisy component images in hyperspectral remote sensing data.

  1. The manuscript studies the analysis of optimal operation points on AVIF and HEIF encoder compression for remotely sensed images with additive white Gaussian noise. The manuscript has some limitations, e.g., only single-channel images are studied for remotely sensed images, only additive Gaussian white noise is studied for noisy images, and five editors are studied for compression. Is it still generalizable to other forms of remotely sensed imagery, other forms of noisy imagery, and other forms of compression?

To our opinion, the proposed approach is generalizable to other forms of RS images as well as medical images. We have already given an example how to deal with compression of component image contaminated by signal-dependent noise (see an example and additional explanation in Section 4 of the revised paper). The use of proper variance stabilizing transforms before compression and after decompression can be the general option for any case of signal-dependent noise (e.g., pure multiplicative). In fact, we (mostly, PhD student B. Kovalenko) spent three years in trials to understand how to compress single channel and three-channel noisy images by the BPG encoder (see, e.g., https://www.scopus.com/authid/detail.uri?authorId=57222125997). However, to the best of our knowledge, such analogs (counterparts) of BPG as HEIF and AVIF have not been studied for the case of noisy image compression. So, this paper is our attempt to get initial imagination of what can be done for HEIF and AVIF. Since many similarities have been found, we expect to continue the research with producing more general results and recommendations for HEIF and AVIF. Meanwhile, we also have to stress two things. First, each coder has its own parameter that controls compression and this parameter can be “convenient” for solving some tasks and “inconvenient” for solving other tasks. This relates to QF for AVIF and HEIF. Second, the coders BPG, AVIF, and HEIF have been originally designed for color images (videos) and it is still necessary to decide how to modify them for applying effectively for multichannel RS images. Some of this aspects are reflected in pages 15 and 16 of the revised paper.         

  1. What is the specific information about the two remote sensing images involved in the experiment, Frisco and Diego? Are they panchromatic images or single-channel images obtained after multispectral image processing or others? What are their sources?

The images are taken from https://sipi.usc.edu/database/database.php?volume=aerials where they are color images of tiff format. This information has been added to text (page 3).   

  1. The description of the techniques associated with the experiments and the analysis of the experiments are mixed, which greatly reduces the readability of the manuscript.

We have tried to give more explanations and present the materials clearer. In particular, we have rewritten the beginning of Section 3. Please also keep in mind that we had several stages of experiments intended on: 1) showing that optimal operation point is possible for AVIF and HEIF coders; 2) comparison of performance of five encoders; 3) analysis of optimal quality factor values for AVIF and HEIF; 4) showing how to reach optimal operation point for them; 5) showing an example for real-life image contaminated by signal-dependent noise; 6) discussing other questions like computation time and so on.    

  1. An overall flow diagram is required to enhance understandability.

Thanks for this proposition. We have added the flow diagram in Fig. 18 in the revised paper. Some explanations are given in pages 15 and 16. 

  1. Too many outdated references are cited and it is recommended to look for more recent and cutting-edge references.

Citing “old” references, we tried to show the history of development of research in the area of lossy compression of noisy images. We have added 8 new references dated 2016-2024 (references ## 10, 11, 22, 39-41, 57, and 58 in the revised version). 

  1. The analysis for the additive white Gaussian noise is a good choice for the study due to its commonality in practical situations. However, the manuscript should mention if other types of noise will also be considered or if there are plans for future work in this area.

We agree with this comment. Our answer is already given above in item 2. Also note that the cases of signal-dependent noise have been already studied by us (mostly, by our co-author PhD student B. Kovalenko) for the BPG encoder (https://www.scopus.com/authid/detail.uri?authorId=57222125997). So, we hope to carry similar research with HEIF and AVIF in the future. This is mentioned in Conclusions.

Round 2

Reviewer 1 Report

Comments and Suggestions for Authors

The questions raised have been revised and agreed to be published.

Author Response

We would like to thank the distinguished reviewer for his valuable contribution to our publication. Authors

Reviewer 2 Report

Comments and Suggestions for Authors

The authors have made a lot of efforts to improve the quality of the manuscript, and the quality of the paper has been greatly improved.  It is recommended for publication now .

Comments on the Quality of English Language

Minor editing of English language required.

Author Response

(The authors gave the same response as above.)
